# Becoming aWARE: The Development of a Web-Based Tool for Autism Research and the Environment

Anisha Singh [1], Cindy P. Lawler [1], Vickie R. Walker [1], Katherine E. Pelch [2], Amanda E. Garton [1], Andrew A. Rooney [1] and Astrid C. Haugen [1,*]

1   National Institute of Environmental Health Sciences, Durham, NC 27709, USA; anisha.singh3@nih.gov (A.S.); cindy.lawler@nih.gov (C.P.L.); vickie.walker@nih.gov (V.R.W.); amanda.garton@nih.gov (A.E.G.); andrew.rooney@nih.gov (A.A.R.)
2   Natural Resources Defense Council, San Francisco, CA 94104, USA; kpelch@nrdc.org
*   Correspondence: haugen@niehs.nih.gov

**Abstract:** A sharp rise in autism spectrum disorder (ASD) prevalence estimates, beginning in the 1990s, suggested factors additional to genetics were at play. This stimulated increased research investment in nongenetic factors, including the study of environmental chemical exposures, diet, nutrition, lifestyle, social factors, and maternal medical conditions. Consequently, both peer- and non-peer-reviewed bodies of evidence investigating environmental contributors to ASD etiology have grown significantly. The heterogeneity in the design and conduct of this research results in an inconclusive and unwieldy 'virtual stack' of publications. We propose to develop **a W**eb-based tool for **A**utism **R**esearch and the **E**nvironment (aWARE) to comprehensively aggregate and assess these highly variable and often conflicting data. The interactive aWARE tool will use an approach for the development of systematic evidence maps (SEMs) to identify and display all available relevant published evidence, enabling users to explore multiple research questions within the scope of the SEM. Throughout tool development, listening sessions and workshops will be used to seek perspectives from the broader autism community. New evidence will be indexed in the tool annually, which will serve as a living resource to investigate the association between environmental factors and ASD.

**Keywords:** autism; autism spectrum disorder (ASD); environmental factors; environmental pollutants; systematic evidence map (SEM)

## 1. Introduction

The sharp increase in prevalence estimates for autism spectrum disorder (ASD), observed beginning in the late 1990s, attracted public attention and sparked greater federal funding for autism research. This dramatic rise in ASD rates further helped to catalyze interest in the potential contribution of environmental factors because it seemed inconsistent with being solely genetic [1,2]. Significant advances in understanding the complex genetic architecture relevant to ASD were made during the past twenty-five years because of increased research investments [3], while initial progress in research on environmental contributors was slower. Nevertheless, there is now a large and rapidly growing body of literature reporting the association of ASD with early developmental exposure to environmental chemicals such as air pollutants, pesticides, heavy metals, and endocrine active compounds [4–6]. The understanding of ASD etiology in the scientific community now reflects multifactorial causation, with interactions among genes, environmental chemical exposures, and other nongenetic factors, including diet, nutrition, lifestyle, social factors, and maternal medical conditions [7–9]. Given the complexity and size of the literature on ASD and environmental factors, it is challenging to find or synthesize all the relevant studies for questions on their potential association. Therefore, we propose to develop a curated and regularly updated Web-based tool for Autism Research and the Environment (aWARE).

Bibliographic search engines such as PubMed, Scopus, and Google Scholar provide relatively simple means to search for scientific articles on ASD and the environment. Despite the unprecedented levels of access that these tools provide, scientists and other interested members of the ASD community (e.g., affected individuals/families and advocacy organizations) are faced with two challenges: (1) finding all the studies relevant to a given question, and (2) keeping up with the pace of ASD environmental research publications (i.e., ~1000/year in 2021 and 2022) against a backdrop of explosive growth in the overall body of autism research. Figure 1a shows this growth and that autism environmental research parallels the increase in both autism prevalence and all autism research [10]. Furthermore, the rapid growth of human and nonhuman research in autism and the environment, a subset of the larger body of research in autism, since the year 2000, is shown in Figure 1b. This literature is considerably heterogeneous in its study parameters, from the categories of environmental exposure to methodologies (e.g., epidemiological, clinical, nonhuman experimental) and measured endpoints (e.g., ASD diagnosis, molecular or cellular alterations). Commentaries, systematic reviews, and meta-analyses provide summaries and syntheses of portions of the literature to address specific questions; however, these approaches are not designed to capture the literature more broadly, or to be readily updated to track research on an entire field of study. Improving health communication and community access to the large amounts of information on ASD and environmental contributors will improve health and science literacy as well as facilitate better (i.e., evidence-based) research questions by scientists and potential future interventions considered by clinicians and their patients.

The broader ASD community is engaged by disseminated research in various ways, including online portals providing digital access to full-text primary journal articles, healthcare and service providers, mainstream media, advocacy and government websites, online support groups, blogs, and social media [11–13]. Research findings about the association of an environmental exposure with ASD garner temporary media attention, but often fail to place these findings in the larger context of prior research, or to provide clear health communication messages. Studies assessing online sources of information on ASD report that the data or information provided are of varying quality and often fail to cite primary source evidence [13–16].

The aWARE database that we propose responds to the high levels of broad-based community interest and need, the growing body of research, and the limitations of existing information sources. It will be curated and updated regularly with information about environmental health studies on ASD and relevant endpoints. The main objective of the first tool deployment is for interested parties to have access to a reliable resource, not only to comprehensively identify environmental ASD research but also to help navigate conflicting and inconclusive research information [13,14,16]. Initially, we will focus on contributors to ASD in the category of environmental chemicals and pollutants. As the aWARE database evolves, there may be room to expand and consider broader categories of environmental factors. The tool will enable users to search, sort, and filter studies on ASD and environmental factors in a way that is user-friendly and interactive, provides data visualizations, facilitates the summarizing and integration of the literature, and helps to identify knowledge gaps.

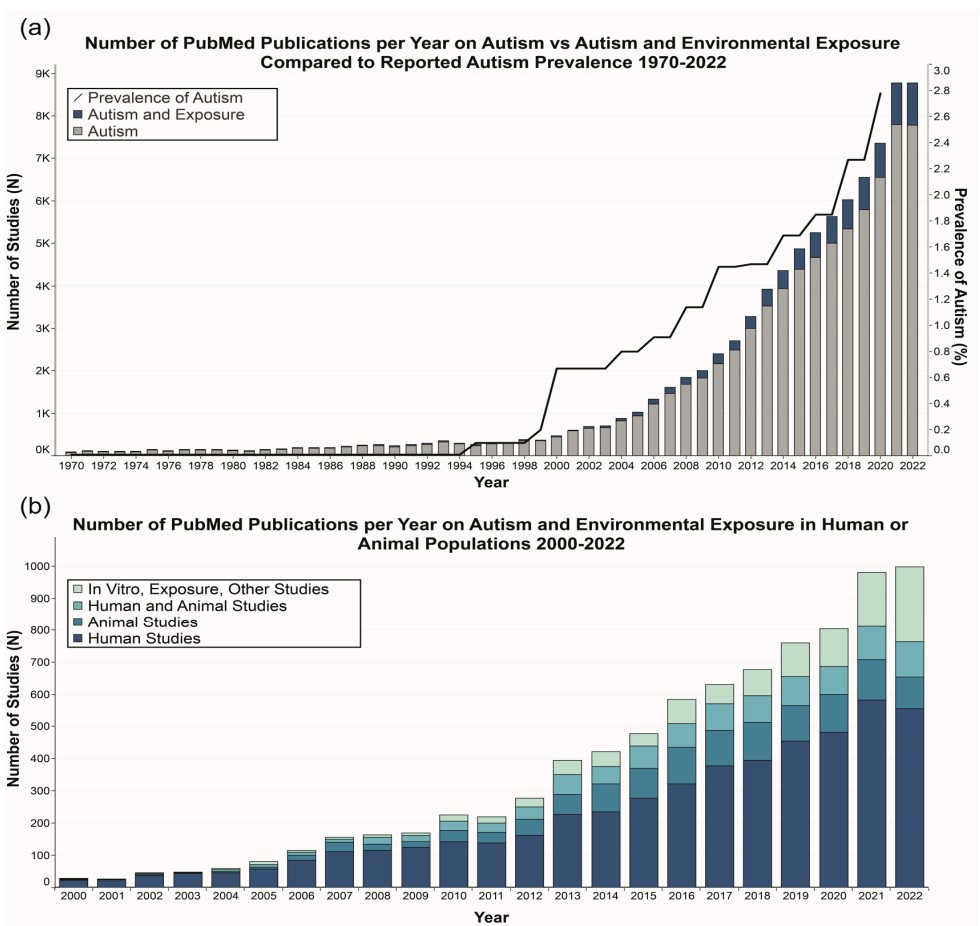

**Figure 1.** The increase in autism prevalence, ASD publications, and ASD environment publications over time. (**a**) The number of ASD publications in PubMed by year with ASD studies and ASD/environment publications displayed in different shades. (**b**) The number of ASD/environment publications in PubMed by year with human, non-human animal, and other studies (including in vitro or exposure studies as well reviews or editorials) displayed in different shades. Autism prevalence is displayed in (**b**) without 2021 or 2022 because it is not yet calculated for those years as of the date of this manuscript. The number of publications is reported at a high level using a simple literature search [PubMed database only using search query: (autism OR autistic OR "Autistic Disorder"[Mesh] OR "Autism Spectrum Disorder"[Mesh]) AND (Environment OR "Environmental Pollution"[Mesh] OR Exposure)]. Note that a more detailed literature search, or the inclusion of additional databases, would identify more studies.

## 2. Role of Evidence Mapping in Public Health and the aWARE Tool

This paper outlines a research effort to establish an interactive database (aWARE) developed with systematic evidence mapping, a methodologically rigorous and transparent process that has emerged in recent years for the collection and display of information on complex topics into systematic evidence maps (SEMs).

### 2.1. How SEM Can Support Communication and Decision Making for ASD

SEMs combine visual presentation with a methodologically rigorous approach adapted from the social sciences to synthesize heterogenous evidence in a transparent and reproducible fashion. These methods are increasingly used in public health because they are well suited to identify and link the broad range of data relevant in addressing the complexity of evidence typically required to answer environmental health questions. The core purpose of an SEM is to provide a comprehensive understanding of the evidence base through interactive outputs [17]. Similar to a systematic review, the SEM process starts with prob-

lem formulation and the development of a specific objective, or research question, and the evaluation protocol. The protocol outlines the stepwise approach to gathering information to answer this research question—a comprehensive literature search, screening for relevant studies, and data extraction/evidence capture. However, unlike systematic reviews that synthesize evidence on relatively narrow questions, SEMs identify, characterize, and present information on broad and often complex research questions. Since SEMs do not synthesize evidence, the information presented can address multiple research questions within the scope of the SEM.

The potential associations between environmental exposures and ASD are exactly the kind of broad and complex research questions that will benefit from an SEM that can be used to promote greater awareness and understanding of the available research. We propose to follow the first three steps of the OHAT systematic review framework developed by the Division of Translational Toxicology and National Toxicology Program to conduct systematic evidence mapping—problem formulation and planning, searching and screening for relevant studies, and a limited data extraction step [18]. The problem formulation and planning are informed by previous efforts to map the ASD and environment literature [6]. This SEM will serve as the basis for the development of the aWARE tool, which will be a National Institute of Environmental Health Sciences (NIEHS)-hosted, accessible, and interactive web-based tool for autism research and the environment. The aWARE tool will have the transparency and scientific rigor of systematic review methods. The SEM will capture and display study attributes from included studies, allowing studies to be categorized and then searched, sorted, or filtered based on key concepts (Table 1). These concepts include environmental exposures, ASD assessment methods, study type (animal/human), and study design (e.g., experimental, cohort, cross-sectional), with additional factors identified based on feedback from the ASD interest community.

The aWARE tool will follow best practices for SEMs, such as maintaining data in native format and categorizing studies by key concepts. The comprehensive approach enables users to efficiently identify evidence gaps and evidence clusters across environmental ASD research as a whole or by narrow topics such as specific chemical exposures. Consequently, captured data would be curated in alignment with the flexible user-driven criteria to ensure maximum transparency and facilitate users' ability to compare studies within a conflicting and heterogenous evidence base [17]. In contrast to the SEM by Pelch et al. [6], which provides a snapshot of the evidence at a fixed point in time, we plan to update the database annually and develop an "evidence surveillance" workflow to automate the identification of new studies. Newly available information that is relevant will be extracted and indexed within the evidence map so that the tool can serve as a living resource. Thus, the aWARE tool will facilitate an understanding of the current state of the science on potential environmental influences on ASD and help to monitor concerns and emerging trends over time [17].

### 2.2. Importance of Autism Community Engagement

For greater impact, it is critical that ASD and environment research is accessible to, and informed by, the full range of consumers of this information. Therefore, we conducted four listening sessions in 2022–2023. These sessions were small, targeted meetings with affected individuals, parents, scientists, clinicians/physicians, and autism advocacy groups. The main purpose of these listening sessions was to develop an understanding of features that would increase the likelihood of the use and utility of this tool by the autism interest community. This active community engagement forms the foundation of this tool, which will facilitate community trust-building and lead to the enhanced use of the tool.

**Table 1.** Examples of key concepts anticipated to be captured in the aWARE SEM.

| Key Concepts | Example Broad Categories | Example Narrow Categories or Individual Factors |
|---|---|---|
| Exposures | Air pollutants<br>Pesticides/herbicides<br>Personal care products<br>Flame retardants | Individual chemicals<br>Individual chemicals<br>Individual chemicals<br>Individual chemicals |
| ASD-related outcomes | ASD diagnosis (humans) | Specific diagnostic/screening tools (ADI-R, ADOS)<br>ASD-related endpoints evaluated using ICD/DSM criteria |
| | ASD-related endpoints (experimental animals) | Reciprocal social communicative behavior<br>Classic stereotyped and repetitive behavior |
| Evidence stream and study design | Humans | Longitudinal study<br>Case–control study<br>Cross-sectional study |
| | Experimental animals | Non-human primates<br>Rodents<br>Zebrafish<br>*Caenorhabditis elegans*<br>*Drosophila melanogaster* |
| Study quality-related details | Selection details | Selection, allocation, randomization, etc. |
| | Attrition details | Attrition or exclusion from analysis |
| | Performance details | Experimental conditions, blinding during study |
| | Detection details | Information on blinding, exposure characterization, outcome assessment, etc. |
| | Confounding details | Consideration of confounding in study design or analysis |
| | Reporting or other details | Selective reporting or other |

ADI-R: Autism Diagnostic Interview-Revised, ADOS: Autism Diagnostic Observation Schedule, ICD: International Statistical Classification of Diseases and Related Health Problems Criteria, DSM: Diagnostic and Statistical Manual of Mental Disorders. Note: This table only includes select examples and is not intended to be complete (i.e., does not cover all exposure categories, does not specify all ASD-related outcomes). It illustrates key concepts that we anticipate capturing from individual studies for the SEM. The specific factors selected will be determined based on input from advisors and input from the autism community. Major categories such as environmental exposures, ASD-related outcomes, evidence streams, and study quality related details are planned. The SEM will collect information in a tiered fashion such that evidence maps can show or hide more details. Using the exposure category as an example, a user could search for, or display, all studies that investigate a potential association with "air pollution" or a specific air pollution chemical (e.g., particulate matter or PM).

## 3. aWARE—A Resource for the Autism Interest Community

The interactive and user-friendly nature of aWARE is the key innovative feature underlying the value of this curated database. Based on the specific research interest, users will have the option to search, sort, explore, select, and download a specific subset of studies on environmental contributors to ASD. Some examples of how distinct users could use this tool are summarized as follows. (1) Autistic individuals and their families will achieve awareness of the breadth of research and a deeper understanding of contributing environmental agents. (2) For advocacy groups and funders, this tool will provide the evidence base to advocate for research and evidence-based regulatory policies. (3) For clinicians, aWARE will provide a means to keep abreast of environmental autism research and address patient questions with reliable scientific information. (4) Finally, by identifying evidence gaps and evidence clusters, this tool will equip scientific researchers with a comprehensive resource to support the formulation of focused research questions and study design. We anticipate that the tool will enable researchers to readily identify studies for evidence integration; the tool will further facilitate the conduct of meta-analyses of exposures of interest and the examination of pooled estimates to make informed conclusions and recommendations.

The aWARE project involves five phases, as shown in Figure 2, incorporating multiple points for engagement and feedback from the autism interest community: (1) planning, initial autism community engagement, and SEM protocol development; (2) developing draft SEM and visuals for input from the autism interest community; (3) completing the SEM and web-based platform and beta testing; (4) launching of the web-based tool; and (5) annually updating the SEM and refining aWARE based on user community input. At present, we are in the first phase of the project, and the team at NIEHS is working with external technical experts and autism community members as well as incorporating "lessons learned" from Pelch et al. [6] to fine-tune the SEM protocol. Following the best practices of systematic review methodology, the protocol will be peer-reviewed and published, assuring transparency and rigor in the approach, from the literature search to data curation. We will use a structured and tiered approach for information extraction focused on a queryable data format. Opportunities to implement automated, or semi-automated, approaches, such as the Data EXTRaction (DEXTR) tool [19] or other platforms, will be pursued to supplement established approaches for information extraction and updating the database. Although annual literature searches are planned, we will update more frequently if automated workflows can be established to increase efficiency in the process. An initial data extraction on a subset of relevant ASD studies will be used to prepare draft visuals for aWARE. These will be presented at a focused workshop, so that a broad range of community perspectives can be integrated during further development of the tool. Additional refinement of the tool and visuals will be done during internal and external beta testing prior to the launch of aWARE. Continued engagement from the autism interest community, throughout the development and deployment of aWARE, will ensure that not only is the first version of the tool successful, but that future updates will only enhance its functionality.

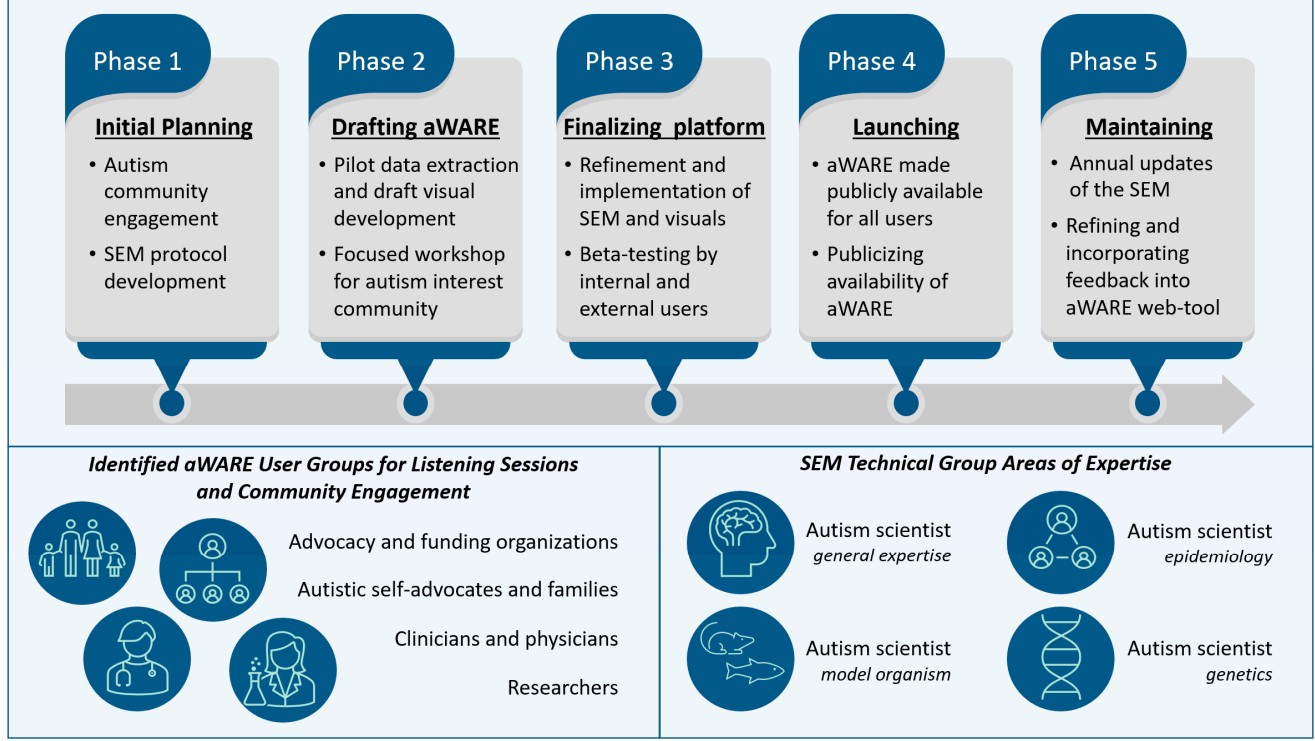

**Figure 2.** Phases and importance of engaging the autism community and external experts in development and maintenance of the aWARE tool.

## 4. Conclusions

Given the large and varied body of research on ASD, it is challenging to find or synthesize all the relevant studies for questions on the potential association between environmental exposures and ASD. Researchers, funding agencies, physicians, advocacy

organizations, parents, affected individuals, and other members of the public need a curated, user-friendly resource that identifies studies and presents relevant data to inform research and public health decisions on ASD. This need is only going to become more challenging as the field attracts growing research interest and more publications. Therefore, we have begun to develop the aWARE tool, a curated, interactive, user-friendly, web-based SEM that identifies and aggregates the broad and diverse scientific research on ASD and the environment. This interactive approach will enable users to search, sort, or filter studies in the aWARE tool for specific questions (e.g., exposures); identify data-poor areas to direct research or data-rich areas to focus synthesis; and explore emerging evidence and new avenues of outcome assessment. Through deliberate engagement with the autism interest community, the aWARE tool can play an important role in laying the foundation for evidence-based decision making and a greater understanding of the complex association between the environment and ASD.

**Author Contributions:** Conceptualization, C.P.L., A.A.R., A.C.H., A.S. and A.E.G.; methodology, A.S., C.P.L., V.R.W., K.E.P., A.E.G., A.A.R. and A.C.H.; writing—original draft preparation, A.S., A.C.H., C.P.L. and A.A.R.; writing—review and editing, A.S., A.A.R., C.P.L., A.C.H., A.E.G., K.E.P. and V.R.W. All authors have read and agreed to the published version of the manuscript.

**Funding:** Anisha Singh was supported through an NIEHS visiting fellowship. ICF support was provided by the Intramural Research Program (ES103379-01) at NIEHS, National Institutes of Health, and was performed for NIEHS under contract GS00Q14OADU417 (Order No. HHSN273201600015U).

**Institutional Review Board Statement:** Not applicable.

**Informed Consent Statement:** Not applicable.

**Data Availability Statement:** Data will be available in a publicly accessible repository upon project completion.

**Acknowledgments:** We thank Kimberly Gray and Brandiese Beverly for the helpful review and comments on a draft manuscript. We appreciate the valuable insights and comments provided by the listening session participants. We also thank Kelly Shipkowski, the NIEHS project contract officer representative, for contract advice and management; and Jo Rochester, Robyn Blain, Anthony Hannani, and Angelina Guiducci from ICF, contractor to NIEHS, for their contributions to problem formulation and figure development for the manuscript.

**Conflicts of Interest:** The authors declare no conflict of interest.

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
