# Peer review of "Becoming aWARE: The Development of a Web-Based Tool for Autism Research and the Environment"

_jox, doi:10.3390/jox13030031_

Round 1
Reviewer 1 Report
This short communication shows a project to be performed, that appears to be interesting.
As minor issue, my question is if annually updating the SEM is inadequate and more updates will be needed.
English is fine.
Author Response
Reviewer 1:
Comments and Suggestions for Authors
This short communication shows a project to be performed, that appears to be interesting.
As minor issue, my question is if annually updating the SEM is inadequate and more updates will be needed.
Authors: Thank you for your comment. We added text to lines 210-213 (track changes version) to indicate that we will incorporate more frequent literature searches and updates to the database if automated workflows can be established to increase efficiency in the process.
Reviewer 2 Report
The manuscript by Singh and colleagues addresses a very pertinent topic, given the increase in ASD prevalence and the need to develop tools to facilitate the search and systematization of the heterogeneous literature available on ASD and environmental factors.
I believe that this paper fits the scope of the Journal of Xenobiotics, and suggest only the following:
- How exactly will the aWARE database “be curated and updated regularly with information about environmental health studies on ASD and relevant endpoints” (lines 90-91)? The manuscript mentions annual updates, but maybe more information could be added on this.
- Maybe the five development phases of the aWARE project, as well as the main groups of beneficiaries from this tool, could be summarized in the form of a figure.
Author Response
Reviewer 2:
Comments and Suggestions for Authors
The manuscript by Singh and colleagues addresses a very pertinent topic, given the increase in ASD prevalence and the need to develop tools to facilitate the search and systematization of the heterogeneous literature available on ASD and environmental factors.
I believe that this paper fits the scope of the Journal of Xenobiotics, and suggest only the following:
- How exactly will the aWARE database “be curated and updated regularly with information about environmental health studies on ASD and relevant endpoints” (lines 90-91)? The manuscript mentions annual updates, but maybe more information could be added on this.
Authors: Thank you for your comment. The text in question introduces the approach for developing the aWARE tool at a high level in lines 92-104 (track changes version) with detailed explanation of the methods immediately following in lines 110 – 168 (track changes version). We feel that the current text appropriately introduces the topic before going into greater detail.
We edited the text in the lines 210-213 (track changes version – text included below) to increase clarity about annual updates.
“Opportunities for implementing automated, or semi-automated, approaches, such as the Data EXTRaction (DEXTR) tool [19] or other platforms, will be pursued to supplement established approaches for information extraction and updating the database. Although annual literature searches are planned, we will update more frequently if automated workflows can be established to increase efficiency in the process".
- Maybe the five development phases of the aWARE project, as well as the main groups of beneficiaries from this tool, could be summarized in the form of a figure.
Authors: Thank you for this suggestion. We have added a new figure (see Figure 2) to address this suggestion.